# Standardization and Key Aspects of the Development of Whole Yeast Cell Vaccines

**DOI:** 10.3390/pharmaceutics14122792

**Published:** 2022-12-14

**Authors:** Anna Jéssica Duarte Silva, Crislaine Kelly da Silva Rocha, Antonio Carlos de Freitas

**Affiliations:** 1Laboratory of Molecular Studies and Experimental Therapy—LEMTE, Department of Genetics, Federal University of Pernambuco, Recife 50670-901, Brazil; 2Genome Laboratory, Department of Biology, Federal Rural University of Pernambuco, Recife 52171-900, Brazil

**Keywords:** whole cell vaccine, antigenic delivery, biotechnology, vaccine improvement, yeast-based vaccines

## Abstract

In the context of vaccine development, improving antigenic presentation is critical for the activation of specific immune responses and the success of immunization, in addition to selecting an appropriate target. In this sense, different strategies have been developed and improved. Among them is the use of yeast cells as vehicles for the delivery of recombinant antigens. These vaccines, named whole yeast vaccines (WYVs), can induce humoral and cellular immune responses, with the additional advantage of dispensing with the use of adjuvants due to the immunostimulatory properties of their cell wall components. However, there are some gaps in the methodologies for obtaining and validating recombinant strains and vaccine formulations. The standardization of these parameters is an important factor for WYVs approval by regulatory agencies and, consequently, their licensing. This review aimed to provide an overview of the main parameters to consider when developing a yeast-based vaccine, addressing some available tools, and highlighting the main variables that can influence the vaccine production process.

## 1. Introduction

Biotechnologically relevant yeasts, such as *Saccharomyces cerevisiae* and *Pichia pastoris*, are recognized expression systems used to produce recombinant proteins derived from animals, plants, bacteria, fungi, and viruses [1]. Currently, it is estimated that yeast produces approximately 20% of the biopharmaceuticals available on the market [2,3]. Among the products are vaccine antigens used for prevention and therapy against different infectious agents, especially viruses such as Hepatitis B, SARS-CoV-2, and Human Papillomavirus [4,5,6,7]. These microorganisms can be engineered to obtain high levels of recombinant proteins, given the available knowledge about their physiological and genetic characteristics [8,9].

Several studies in the last 20 years have demonstrated the possibility of using these yeasts for purposes other than biofactories, such as vaccine delivery systems in strategies designed for both human and veterinary applications (Figure 1) [10,11]. The GRAS (“Generally Recognized As Safe”) status, as well as the immunostimulatory properties combined with the ability to favor specific immune responses to the carried antigen, are some of the main advantages of this platform [1,12]. The yeast-associated immune response is mediated by D-glucose polysaccharides (-glucans), mannose polymers covalently linked to peptides (mannoproteins), and *N*-acetyl-b-D-polymers glucosamine (chitin), which comprise the cell wall. These structural polymers are recognized as Pathogen Associated Molecular Patterns (PAMPs) and have immunostimulatory properties reported in several studies involving vaccine strategies. Thus, the yeast cell can be used as a vehicle protector of vaccine antigens and as an adjuvant [10,13,14,15].

Whole proteins or epitopes of these proteins, critical for inducing the immune response against a target pathogen, can be carried as vaccine antigens in the intracellular compartment or exposed on the cell surface of yeasts via surface display systems [16,17,18]. Furthermore, nucleic acid delivery for DNA and mRNA vaccines has been evaluated, in addition to the delivery of interference RNAs (siRNA) used for immunomodulation in therapeutic approaches [19,20,21].

Despite the advances and promising results of this vaccine approach, including successful clinical trials [22,23,24], there are still some gaps concerning the methodologies applied in the validation processes and obtainment of whole yeast vaccines. This review aims to provide a comprehensive snapshot of parameters and tools to consider during WYV development. In addition to the antigen selection issue, some critical aspects must be considered for the standardization and establishment of this platform (Figure 2). These aspects include dose measurement, scaling-up, routes of administration, inactivation methods, and time and temperature stability analyses. The validation of these processes can contribute to the approval by regulatory agencies and the consequent licensing of prophylactic and therapeutic WYVs.

## 2. Yeast Genera

The discussions in this article are related to the main yeast genera of biotechnological interest: *S. cerevisiae*, *Pichia pastoris* (now named *Komagataella phaffii*), *Hansenula polymorpha*, *Yarrowia lipolytica*, *Schizosaccharomyces pombe*, and *Kluyveromyces lactis*. The choice of the yeast species used as a host cell can interfere not only with the production of recombinant proteins but also with the vaccine-induced immune response [25,26]. Bazan et al. (2018) observed differences in the level of stimulus provoked for the activation of dendritic cells concerning the expression of surface markers such as CD40, CD58, CD80, CD83, and CD86 and the cytokines released by these cells [27]. These differences were observed between distinct species and among strains of the same species.

It is essential to know the profile of the immune response induced by the yeast chosen as a carrier and to understand the immunological properties that characterize it as an adjuvant. The cell wall of yeast is the main element responsible for its adjuvant activity. However, it is worth mentioning that this organelle is not static, and its architecture can undergo dynamic changes depending on growth conditions as well as culture media and also vary according to the yeast genus or species used [28]. The proportion and arrangement of the cell wall components of each species can influence yeast recognition by the immune system and the uptake by antigen-presenting cells (APCs) [25]. While the distribution of mannan polymers appears to be homogeneous among different genera such as *S. cerevisiae*, *P. pastoris*, *Sz. pombe* and *K. lactis*, the positioning of β-glucans can differ and show variated patterns between budding and fission yeasts [15]. Furthermore, depending on the yeast used, the tools available for optimizing the delivery system may differ. Some species, such as *S. cerevisiae* and *P. pastoris*, have a broader set of vectors, well-characterized promoters, and knowledge about the best culture conditions for the expression of vaccine antigens [8].

## 3. Yeast Cell Inactivation

The yeasts used in the WYV are neither pathogenic nor toxic. Despite this, they are usually inactivated before immunization to minimize eventual risks regarding vaccine biosafety. The aims of yeast inactivation are reproductive capacity loss and the elimination of cell metabolic activities. The employed protocols are simpler than those used for the inactivation of viruses or bacteria because they do not require chemical processes. Yeast inactivation involves heat treatment, starting with incubation at temperatures ranging from 56–95 °C (Table 1), which is sometimes accompanied by lyophilization (often freeze-drying). Determining the most suitable heat temperature can be influenced by several factors, from cell concentration to the heating method or yeast species used [29]. The efficiency of this procedure can be evaluated through viability tests that include visualization of the yeasts under an optical microscope after incubation with vital dyes such as methylene blue and trypan blue or seeding the treated yeasts in a suitable culture medium for growth evaluation [30].

Despite affecting the viability of yeasts in terms of their reproductive capacity, different studies have found no effect on vaccine antigen stability and functionality or yeast immunoreactivity concerning their recognition and uptake by APCs [15,31]. Although there may be differences due to the cellular architecture of the chosen yeast species, it has been suggested that heat treatment may favor the exposure of the β-1,3-glucan layer on the cell surface. Once this is one of the most immunogenic components of the yeast wall, this exposure may facilitate the binding to receptors such as Dectin-1 present in APCs and M cells [15].

In the process of preparing and standardizing doses, it is important to include the verification of protein levels present in the yeast cell suspension after heat treatment. This evaluation can be performed through SDS-PAGE and western blot, as executed by Arnould et al. 2012 who observed that the recombinant protein remained intact after the inactivation procedures [32]. Similarly, Kumar and Kharbikar (2021) observed that the protein levels remained stable through freeze (−80 °C)/thaw cycles and lyophilization [33]. Further studies should evaluate how long WYVs can be stored without losing stability and efficiency, and which temperatures can be adopted.

Although it is still understudied, the influence of heat treatment or the difference between administering heat-killed or live WYV can be related to the administration pathway adopted in the vaccination schedule. Capilla et al. (2009) observed differences in vaccine-induced responses between the intramuscular and oral routes and between live and heat-killed yeast cells administered orally [34]. In this study, oral immunization with live yeast induced higher protection after an immunological challenge than the HKY vaccine (heat-killed yeasts). Several factors could influence this result, including the inactivation protocol, the dose concentration, and the type of delivery (intracellular or surface displayed). Moreover, after heat inactivation and lyophilization, the yeast cells lose their replicative capacity and classification as an organism (Directive 2001/18/EC). This can also change their classification as genetically modified organisms (GMO), which can facilitate adherence to the WYV and simplify their licensing [29,35].

## 4. Storage and Stability of Vaccine Preparations

Vaccine stability is critical in the process of distributing these immunogens to the population. The conditions of maintenance and storage can directly influence vaccine stability and its efficiency as an immunogen [33]. One of the most alarming problems affecting the distribution and storage of vaccines currently produced is the need to keep these immunoreagents in cold chains [36]. However, this requirement increases the cost associated with the vaccine product, which can reach 80%, and can be a bottleneck in vaccine acquisition by underdeveloped countries that do not have the necessary infrastructure [36,37]. Therefore, measures that increase the stability of vaccines at room temperature or even new approaches that escape the need for continuous refrigeration would positively impact the mass vaccination of the population [38]. Yeast-based vaccines are a promising alternative in this scenario.

Kumar (2018) observed that the yeast cells remained stable both under refrigeration (2–8 °C) and at room temperature (23–25 °C) for one year [39]. Even though the yeasts lost reproductive capacity after heat treatment and long storage at both temperatures, the level of proteins produced in the stationary phase remained stable, especially in the cells kept under refrigeration. This measurement was performed via western blot, using imaging software that calculates the equivalence between the intensities of the bands and protein quantification (Image J Software).

The maintenance of protein levels has been demonstrated in cells in recombinant *P. pastoris*, expressing *Escherichia coli* surface protein, CsgA-GFP, under different conditions of time and temperature: one year at 37 °C, one and a half years at 30 °C, and even after freezing and thawing at −20 or −80 °C [33]. Yeast lyophilization has been identified as a promising procedure for ensuring the storage of yeasts for long periods of time at room temperature until the time of use without compromising antigen integrity or vaccine efficiency [35]. Heat-killed lyophilized WYVs are employed in clinical phase studies [40], where, after adjustment and establishment of doses, the yeasts are bottled in glass vials until the moment of immunization, when they are reconstituted with water for injection in the appropriate volume for the desired dose.

It is worth mentioning that most of the stability and viability studies evaluate the WYV-carrying protein antigens in the intracellular portion. Regarding WYV with surface-displayed antigens, it has already been shown that the vaccine remained viable even after the freeze-drying process. Patterson et al. (2015) observed this property while testing the vaccine in an oral immunization approach in pigs [35]. It is interesting to highlight that the stability of antigens anchored on the surface of yeasts is preserved not only after heat treatment but also after passage through the gastrointestinal tract in strategies that employ oral administration [17]. To date, there are no studies about the stability of yeast-delivered nucleic acid vaccines. Still, it is important to assess whether there are differences in stability according to the antigen and type of carrier.

**Table 1 pharmaceutics-14-02792-t001:** Preclinical studies with whole yeast cells employing different yeast genera, routes, and animal models.

Yeast Genera	Infectious Agent	Inactivation	Route	[Dose]	Animal Models	Ref.
*K. lactis*	Infectious bursal disease virus	90 °C/2 h	SC and oral	SC: 100 µg/200 µL (mouse) and 1 mg/500 µL (chicken). Oral: dried yeast nuggets mixed with feed (end concent. of 5% *w*/*w*).	Mouse and Chicken	[32]
*K. lactis*	Influenza A virus	90 °C/2 h	SC	1, 2 or 5 mg/100 µL	Mouse	[41]
*H. polymorpha*	Hepatitis B	60 °C/2 h	IM	2 × 10^8^ yeast cells/100 µL	Mouse	[16]
*P. pastoris*	*Plasmodium berghei*	60 °C/45 min	SC	30 YU/100 µL	Mouse	[30]
*P. pastoris*	Human Papillomavirus	60 °C/2 h	SC	2.5, 5 and 10 mg	Mouse	[42]
*P. pastoris*	Human Papillomavirus	56 °C/15 min	SC (multipoint injection)	5 µg (2 × 10^2^ yeast cells)	Mouse	[43]
*P. pastoris*	Highly pathogenic avian influenza	-	Oral	6.7 × 10^9^ cells/mL in 2.5 mL	Chicken	[44]
*S. cerevisiae*	Dengue	-	Oral	1.6 g (fresh) in 2.4 mL	Mouse	[45]
*S. cerevisiae*	*Vibrio harveyi*	-	IP	5 × 10^9^ cells/mL in 200 µL	Marine fish (flounder turbot)	[46]
*S. cerevisiae*	*Actinobacillus pleuropneumoniae*	-	Oral	1.5 × 10^9^ cells/day	Mouse	[11]
*S. cerevisiae*	*Porcine circovirus type 2*	(Freeze-dried)	Oral	7 g (freeze dried yeast) in 20 mL of sterile	Pig	[35]
*S. cerevisiae*	H7N9 virus	60 °C/1 h	Oral	150 OD_600_	Mouse	[17]
*S. cerevisiae*	*Coccidioides immitis*	70–75 °C/3 h	Oral, IM	1.2 × 10^8^ cells (5 mg per dose) * 6 × 10^7^ cells (2.5 mg per animal)	Mouse	[34]
*S. cerevisiae*	*Eimeria tenella*	56 °C/1 h and 95 °C/2 min	Oral	1.7 YU in 100 µL and 1.5 × 10^7^ cells/mL (200 µL per animal)	Chicken	[29]

SC: subcutaneous; IM: intramuscular; IP: intraperitoneal. * Live or heat-killed.

## 5. Definition of Doses and Quantification

The unit of measurement of the vaccine doses broadly varies in the different clinical and preclinical studies performed. One of the definitions considers the number of cells per vaccine preparation, adopting “yeast units” (YU), where 1 YU corresponds to 10^7^ cells (approximately equivalent to OD_600_ = 1) or the dry weight in mg or g (1 mg corresponding to approximately OD_600_ = 2) (Table 1). Cell counting can be done in a relative manner by measuring the optical density or by counting through a Neubauer chamber, or using more accurate techniques such as conventional flow cytometry or micro-flow imaging (MFI) [47]. The dry weight is evaluated, in general, after the lyophilization process. There are some concerns when the yeast unit is adopted instead of the dry weight because some protocols that rely on optical density, for example, may not be as accurate as ones that use automated equipment to determine the number of cells and their viability. This eventual imprecision may have an impact on the reproducibility of vaccine trials and the accuracy of dose preparation.

There are variations in the minimum threshold suitable for inducing immune responses and in establishing a concentration of cells per preparation. There is considerable variation between studies that use yeast as vaccines (Table 1), although it is known that there is a correlation between the amount of yeast used and possible immunological effects such as the induction of neutralizing antibodies, for example [16,45]. In clinical studies, a maximum dose of 10 to 12 YU per injection has been adopted, with at least four administrations at different sites. In a clinical trial conducted by Cohn et al. (2019) aimed at patients with tumors with mutations in the Ras oncogene, it was observed that the subcutaneous administration of up to 10 YU did not lead to significant adverse reactions, even with four applications totaling 40 YU [48].

The number and concentration of doses may also differ depending on whether the vaccination is prophylactic or therapeutic. Therapeutic approaches may require more applications to achieve the purpose, which is often related to generating cytotoxic responses and eliminating infected or tumor cells. Preclinical and clinical cancer vaccine studies point to a dose-dependent process where multiple applications seem to lead to an optimal antitumor effect [3]. Multiple-site injections can amplify the immune response by targeting multiple peripheral lymph nodes. Thus, depending on the vaccination strategy, inoculation sites that reach the inguinal, axillary, and subclavicular lymph node beds are chosen [49,50].

Regarding the amount of vaccine antigen per dose, the main criteria are concentration, immunogenicity, and administration route. Some optimizations can help to improve the production of the recombinant antigen and the number of yeast units sufficient to induce protection or treat some pre-existing diseases. Considering the yeast adjuvant properties and, depending on the immunogenicity of the antigen, an adequate immune response can be obtained even with a low concentration of the recombinant protein, only increasing the number of yeast cells per dose [32]. Furthermore, an ideal antigen concentration to reach an optimal dose can be influenced by the route of administration [34]. Oral vaccines may require higher amounts of antigen in terms of concentration or number of cells than parenterally administered vaccines.

The antigen concentration in WYVs, which carry proteins, depends on the yeast species and strain chosen as biofactories and their ability to produce heterologous proteins, as well as the expression vector, which is also subject to optimization. The application of methods capable of detecting and quantifying the presence of the antigen in a defined number of cells includes approaches such as western blot, Yeast ELISA, and flow cytometry [47,51]. Flow cytometry is the most widely used method for quantifying the percentage of positive cells (expressing and exposing the antigen) present in the set of cells that comprise the vaccine dose in WYV that uses the surface display system [35,52].

However, few studies show a correlation between the dose and the actual concentration of the delivered antigen. Bian et al. (2009) quantified the antigen (Hepatitis B virus proteins) and tested concentrations from 0.75 to 1.25 μg in 1 × 10^8^
*H. polymorpha* cells [16]. Similarly, Arnould et al. (2012) assessed the protein concentration per dose through SDS-PAGE and Western blot, estimating a concentration of 0.7 fg of heterologous protein per *K. lactis* cell [32]. The quantification process when the antigen carried is a nucleic acid vaccine is even less explored, so there are no records of well-established methodologies to define the amount of DNA, mRNA, or siRNA carried by the yeast. In these cases, the concentration of the genetic material used to transform the yeast cells can be decisive, but it is important to find out how much has been assimilated by the cell. The most suitable methodologies for these processes may involve conventional PCR to check for the presence of the antigen and qPCR.

## 6. Culture Scaling-Up

Culture and protein production parameters may influence vaccine dose setting and clinical efficacy. The high cell densities achieved in the yeast cultivation process make it possible to obtain thousands of doses per liter of culture. However, some parameters can optimize this production, such as the yeast species used as the host cell, the period established for the culture, the promoter present in the expression vector, and the number of copies of the recombinant DNA per cell [53,54,55]. A fundamental point for the establishment of a biofactory or vaccine platform is the ability and ease of scaling up cultures, starting from pilot protocols and reaching large fermenters for high biomass generation [56]. This ability has been demonstrated for different yeast species such as *S. cerevisiae*, *P. pastoris,* and *Y. lipolytica* and is crucial to indicating the commercial applicability of the vaccine to be produced [54,57].

The employment of bioreactors for biomass generation makes it possible to reach higher levels of expression than from culture flasks commonly used in the first stages of laboratory cultivation [32]. The factors that can be controlled and optimized for better efficiency include culture medium composition, pH control, oxygen availability (dissolved oxygen level), cultivation time, and volume [8].The optimization in the scaling steps involves the choice of the bioreactor, the kinetic features of the yeast, and the mode of operation (continuous, discontinuous, and discontinuous-fed) [54]. In this scaling process, the culture usually starts in the batch process and, further, is adapted to fed-batch, allowing better monitoring and control of nutrient input throughout the production phases [57]. Continuous cultivation has more caveats due to the greater propensity for contamination, mutations in the strains, and instability of the products [54]. Regarding kinetics, it is important to promote a balance between growth and protein synthesis and to pay attention to the fact that this parameter can be affected by the promoter and the number of gene copies. Furthermore, the culture conditions may depend on the study objective, whether it is to recover and purify the antigen or to use recombinant cells as a vaccine vehicle.

The demand for culture scaling up also depends on the immunobiological to be produced. Prophylactic and therapeutic strategies have different requirements regarding the number of doses and individuals to be immunized. Overall, prophylactic vaccines aim at mass vaccination, while therapeutic vaccines comprise more personalized immunotherapy with a smaller target population. Thus, the scale of production of a therapeutic vaccine using yeast as a biofactory and vehicle requires less accumulation of biomass. This particularity of therapeutic approaches even favors nucleic acid-carrying strategies, whose scaling-up procedures can be more complex and are not yet well established.

## 7. WYVs as a Promising Low-Cost Vaccine Platform

The need to develop and improve tools to increase vaccine coverage and promote health equity is becoming increasingly evident, especially in light of the pandemic caused by SARS-CoV-2. Improving vaccine strategies proves to be an interesting work front that can contribute to both efficiency improvements and cost reductions in production and logistics for the storage and distribution of these vaccines. This cost reduction is critical to ensuring coverage in developing countries or where financial resources are limited [58].

Several delivery systems for drugs and vaccines have been studied over the years to improve their efficiency and biodistribution. Some examples include nanocarriers, liposomes, polymeric particles, virus-like particles, and organisms such as viral vectors, bacteria, and yeasts [59,60,61]. Among these, the use of yeasts, especially *S. cerevisiae*, is a more economically viable alternative to other delivery mechanisms previously described [1,62]. Viral vectors are well-established in immunization programs, including for SARS-CoV-2, and are an interesting system to compare with yeast-based vaccines [63]. Despite their immunogenic properties, there are some safety concerns about using viral vectors in vaccines, and they can be neutralized by the host when multiple doses are required [64]. On the other hand, preclinical and clinical studies with WYV show the possibility of repeated immunizations without notification of vector-neutralizing responses [34,50].

WYVs are promising in terms of cost-effectiveness due to several aspects. It is possible to generate recombinant strains from a less complex infrastructure than that used for the cultivation of mammalian cells, insect cells, or even the cultivation of cells for viral propagation (necessary for the production of first-generation vaccines) [56]. The possibility of cultivating yeasts in media that require cheaper inputs is also a facilitating aspect. The thermostability addressed in this review also represents an economic advantage because the non-requirement of refrigeration chains makes it possible to distribute and store the vaccines in more remote locations, in addition to reducing the added value by simplifying the necessary infrastructure.

The most advanced clinical studies using WYVs aim at immunotherapies for chronic diseases such as tuberculosis, hepatitis C, and cancer. Most of these studies comprise the Tarmogen (targeted molecular immunogen) technology, which covers a series of patented strains already tested in preclinical and clinical trials [65,66,67,68]. The findings of these studies are encouraging, demonstrating the safety and viability of this type of vaccine and potentially aiding in the advancement of WYVs developed for prophylaxis.

## 8. Conclusions

The points raised here have practical implications for the development of more effective WYV and can contribute to the reproducibility of procedures. The licensing of a pathogen-specific vaccine or vaccine platform is subject to compliance with the regulations of different agencies. Therefore, all steps involving the development and preparation of vaccine formulations must be well established, characterized, and described to guarantee the quality and safety of the proposed vaccines. WYVs form a relatively recent platform that still lacks some methodological standardization. Among the hallmarks of WYVs obtaining are the precise quantification of the recombinant antigen, the shelf life of the product, storage specifications, and the manufacturing scales. It is essential to have clarity on these aspects as they can impact the understanding of the correlation between dosage and clinical efficacy, in addition to allowing an adequate comparison with conventional vaccine platforms.

## Figures and Tables

**Figure 1 pharmaceutics-14-02792-f001:**
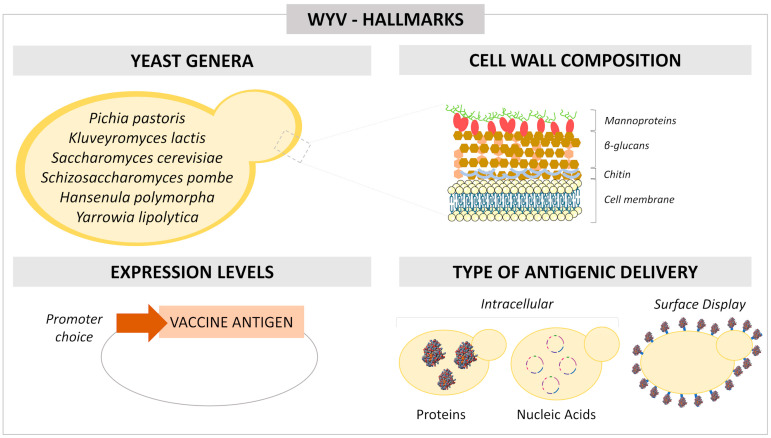
Main hallmarks concerning whole yeast vaccines. These aspects (yeast genera, cell wall composition, target-antigen expression, and the type of antigenic delivery) influence the immune response associated with yeast-based vaccines, as well as their efficacy.

**Figure 2 pharmaceutics-14-02792-f002:**
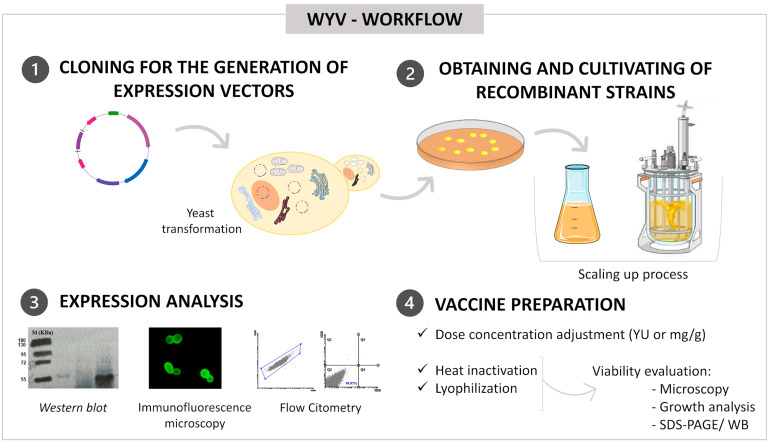
Steps for whole yeast vaccine development that must be performed before immunological assays. Steps for whole yeast vaccine development that must be performed before immunoassays. (1) The first step is antigen selection and cloning it into the appropriate expression vector. (2) The constructed vector will be used to transform yeast cells. After obtaining the strains, cultures are performed in a liquid medium to confirm the expression of the antigen carried and for later preparation of the doses. This cultivation can be done in culture flasks (laboratory scale) or scaled up for production in bioreactors. (3) Different methodologies are used to analyze and confirm the expression levels and antigenic carriage, including western blot, immunofluorescence microscopy, and flow cytometry. (4) In general, yeasts are inactivated by heat and can be lyophilized without altering their viability as a vaccine vector. Some methods can be employed to analyze cell viability, such as microscopy and plate growth. Furthermore, it is important to check the antigen expression levels after the inactivation and dose adjustment processes, through SDS-PAGE and western blot, for example.

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
