# Peer review of "Standardization and Key Aspects of the Development of Whole Yeast Cell Vaccines"

_pharmaceutics, 2022, doi:10.3390/pharmaceutics14122792_

Round 1

Reviewer 1 Report

Review article titled "Padronization and key aspects of the development of whole yeast cell vaccines(pharmaceutics-2085093)" is a nice piece of writing. Authors were able to catpure some of the important issues in development and use of yeast-based vaccines. Issues mentioned in draft were missing from previous articles. I will be happy to see the draft accepted for publication in present journal after author address following minor issues.

1) There is a need for writing improvement and paraphrasing at several places.

2) At several place author use term viability of vaccines, it should be stability or efficacy

3) In draft author mentioned "Another important point regarding heat inactivation and lyophilization is that after these procedures, the yeast cells lose their classification as genetically modified organisms (GMO) which can facilitate the adherence to the WYV as 141 well as simplify their licensing". Author should explain how heat inactivation and lyophilization lead to lose of GMO status. Or modify the statement.

Author Response

We appreciated and considered to manuscript final version all suggestions.

1) There is a need for writing improvement and paraphrasing at several places.

Answer: The text has been revised again for readability. All modifications are highlighted throughout the manuscript.

2) At several place author use term viability of vaccines, it should be stability or efficacy

Answer: To facilitate the analysis of the use of the term “viability”, we have placed below the excerpts where it appears in the text so that you can analyze them individually.

  • “The efficiency of this procedure can be evaluated through viability tests that include visualization of the yeasts under an optical microscope after incubation with vital dyes such as methylene blue and trypan blue or seeding the treated yeasts in a suitable culture medium for growth evaluation “.
  • “Despite affecting the viability of yeasts in terms of their reproductive capacity, (...)”

Answer: Here, viability is related to the reproductive capacity of the yeast cells.

  • “(...) different studies have shown that these treatments do not affect the viability of the vaccine antigen or the immunoreactivity of the yeasts concerning their recognition and uptake by APCs”

Answer: In this case, we replaced viability with stability and functionality of the vaccine antigen.

  • “Further studies should evaluate how long WYVs can be stored without losing viability and efficiency, and which temperatures can be adopted.”

Answer: We exchanged viability by stability.

  • “Even though the yeasts lost viability after heat treatment and long storage at both temperatures, the level of proteins produced in the stationary phase remained stable, especially in the cells kept under refrigeration.”

Answer: We change for reproductive capacity.

  • “It is worth mentioning that most of the stability and viability studies evaluate the WYV carrying protein antigens in the intracellular portion.”

Answer: Here, viability is related to the reproductive capacity of the yeast cells.

  • “There are some concerns when the yeast unit is adopted instead of the dry weight once some protocols that rely on optical density, for example, may not be as accurate as ones that use automated equipment to determine the number of cells and their viability

Answer: In this case, the viability is associated with the maintenance of the cellular structure of the yeasts, mainly after the heat treatment.

3) In draft author mentioned "Another important point regarding heat inactivation and lyophilization is that after these procedures, the yeast cells lose their classification as genetically modified organisms (GMO) which can facilitate the adherence to the WYV as 141 well as simplify their licensing". Author should explain how heat inactivation and lyophilization lead to lose of GMO status. Or modify the statement.

Answer: We appreciate your feedback and updated the text as follows:

“Moreover, after heat inactivation and lyophilization, the yeast cells lose their replicative capacity and classification as an organism (Directive 2001/18/EC). This can also change their classification as genetically modified organisms (GMO) which can facilitate adherence to the WYV and simplify their licensing.”

Reviewer 2 Report

This paper presents an overview of a subject that is not commonly known and could be potentially be a good paper.

1) 'There are some awkward expressions and statements that need to be rephrased:

"it is also explored nucleic acid delivery for DNA and mRNA 57vaccines, besides..."

"Further to the antigen selection issue, some critical..."

The authors may want to check for more awkward sentences.

2 It would help if the authors offer a paragraph or two summarizing the advantages and disadvantages of using yeast vaccines as compared to the conventional and other existing vaccines.

3) One potential complication I can foresee is the complication resulting from introducing foreign (yeast) protein in addition to the antigens. This was waht happened when the SARS-CoV-2 Oxford=Astra-Zaneca vaccine was developed using  Adenovirus. The proteins from the Adenovirus caused rare blood clotting. What is the opinion  of the authors pertaining this issue.

Author Response

We appreciated and considered all the suggestions. The manuscript was uptaded.

1) 'There are some awkward expressions and statements that need to be rephrased:

"it is also explored nucleic acid delivery for DNA and mRNA 57vaccines, besides..." 

"Further to the antigen selection issue, some critical..."

The authors may want to check for more awkward sentences.

Answer: The text has been revised again for readability. All modifications are highlighted throughout the manuscript.

2) It would help if the authors offer a paragraph or two summarizing the advantages and disadvantages of using yeast vaccines as compared to the conventional and other existing vaccines.

3) One potential complication I can foresee is the complication resulting from introducing foreign (yeast) protein in addition to the antigens. This was waht happened when the SARS-CoV-2 Oxford=Astra-Zaneca vaccine was developed using  Adenovirus. The proteins from the Adenovirus caused rare blood clotting. What is the opinion  of the authors pertaining this issue.

Answer to issues 2 -3: We appreciate your suggestions, and to answer the questions above, we have added the topic “WYVs as a promising low-cost vaccine platform”, contextualizing yeast-based vaccines among other antigenic delivery systems. In this topic, we take advantage of the space to address some limitations that systems such as viral vectors have, but that can be circumvented with the use of yeast.

Reviewer 3 Report

The article submitted for review attempts to clearly expose the new opportunity for the creation of vaccines through yeast strains. This already known technique opens a new door to the improvement of the technique. 

I believe that the authors should address at some point in the text the economic benefits, if any, of this technique, which could facilitate the rapid mass production of vaccines in the future. Also if this technique would favor the production of vaccines that are not available at the moment, for example vaccines against cancer cells.

Overall, the article seems to me adequate and conveniently written and I believe that it could be submitted for publication with the proposed modifications.

Author Response

We appreciate the suggestions and have updated the manuscript accordingly.

I believe that the authors should address at some point in the text the economic benefits, if any, of this technique, which could facilitate the rapid mass production of vaccines in the future. Also if this technique would favor the production of vaccines that are not available at the moment, for example vaccines against cancer cells.

Answer: We appreciate your suggestion and, to solve this question, we have added the topic “WYVs as a promising low-cost vaccine platform”.
